# Bayesian Optimal Experimental Design of Streaming Data Incorporating Machine Learning Generated Synthetic Data

**Kentaro Hoffman**
Department of Statistics
University of Washington
Seattle, WA 98101
khoffm3@uw.edu

Tyler H. McCormick
Department of Statistics
University of Washington
Seattle, WA 98101
tylermc@uw.edu

## 1 Introduction

With the rapid advancement of Machine Learning (ML) and Artificial Intelligence (AI), data-based models are quickly becoming an important part of high-profile decision making. For example, ML and AI models are now being used to make medical diagnoses Ahsan et al. [2022], rapidly screen documents Lee et al. [2023], Gongane et al. [2022], determine marketing strategies, Stone et al. [2020], and aid in making parole decisions Wang et al. [2022]. However, in many ways, models are only as good as the data used to train them. Partially due to the success and ease of ML and AI models, there have been concerns that the proliferation of accurate-seeming but inaccurate data such as deepfakes or hallucinations could poison existing sources with lower quality data. The endpoint of this concerning direction is a phenomenon which has been referred to as "model-collapse" Shumailov et al. [2024], Gibney [2024], Gerstgrasser et al. [2024]. Model-collapse refers to a cascade where an AI generates enormous amounts of lower quality synthetic data, which is then used to train a new AI, which in turn generates even lower quality synthetic data due to being trained on lower quality data, and so on. With ML and AI models being more important parts of our decision making process than ever before, it is important to develop new methods to ensure we can get valid statistical inference even in the presence of synthetic data.

To this end, this paper demonstrates two main innovations to aid in statistical inference using synthetic data in dynamic contexts. First, using a class of estimators which give valid statistical inference using synthetic and real data points, even when the operating characteristics of the synthetic data generation process are unknown, we illustrate how to incorporate our proposed estimators into dynamic linear models to analyze streaming data. Second, we combined our proposed estimators with Bayesian optimal experimental design to dynamically determine the optimal ratio of real and synthetic data to minimize model standard error.

## 2 Setup

Borrowing the setup from Szpiro et al. [2010], we posit that, at time $t$, our data is generated from a process is:

$$y(t) \sim N(\phi(x,t), \sigma^2(x,t)) \tag{1}$$

where:

$$\phi(x,t) = \beta(t)x + \Phi^T B_\phi(x) \tag{2}$$

$$log(\sigma(x,t)) = \gamma(t)x + \Psi^T B_\sigma(x). \tag{3}$$

Workshop on Bayesian Decision-making and Uncertainty, 38th Conference on Neural Information Processing Systems (NeurIPS 2024).

Here $B_\phi$ and $B_\sigma$ are vectors of B-splines. We shall say that at time point $T_{-1}$, a group of upstream researchers fit a deterministic ML function $\hat{f}(x)$ (which we shall refer to as the "upstream model") that approximates $\phi(x)$. These researchers are willing to disseminate $\hat{f}(x)$, but do not provide any information beyond this, including sample size or error rates. This mirrors the way modern AI models, such as ChatGPT, are shared.

Later, a different group of researchers is interested in using a combination of real outcomes and upstream-model-generated synthetic outcomes for use in a downstream statistical model. As an example, this could occur in pragmatic clinical trails where one would like to learn the relationship between a clinical outcome, Y, and a biomedical covariate, X. However, if Y is expensive or difficult to collect, then one may use a previously trained prediction model to generate $\hat{f}(X)$ and regress that along with the collected Ys upon X [Gamerman et al., 2019, Williams et al., 2015].

In our paper, our goal will be to describe the linear relationship $\beta(t)$, between $y$ and $x$ at each time step $t \in [0, ..., T]$. Formally, our goal is to estimate $\beta_t | \boldsymbol{Y}^t = \{Y_{real}^t, \hat{Y}_{syn}^t\}$ where $Y_{t,real} = \{y_{1,real}..., y_{t,real}\}$ and $\hat{Y}_{t,syn} = \{\hat{f}(x_{1,syn})..., \hat{f}(x_{t,syn})\}$. A class of estimators which can give parameter estimates using a combination of real and black-box generated synthetic outcomes is known as Inference on Predicted Data (IPD) estimators Hoffman et al. [2024a]. Here, we will use a particular IPD estimator known as Prediction Powered Inference (PPI). In Angelopoulos et al. [2023], the authors demonstrate that one can estimate a linear regression parameter $\beta$ (as well as other convex estimation problems) via a PPI estimator, $\hat{\beta}^{PPI}$

$$\hat{\beta}^{PPI} = \hat{\beta}^{Naive} + \hat{\Delta}$$

where $\hat{\beta}^{Naive}$ is the parameter from linearly regressing $\hat{f}(x_{syn})$ on $x_{syn}$ and $\hat{\Delta}$ is the parameter from linearly regressing $(y_{real} - \hat{f}(x_{real}))$ on $x_{real}$. Angelopoulos et al. [2023] and Angelopoulos et al. [2024] show that these estimators not only give unbiased estimates but can lead to tighter standard errors compared to traditional estimators.

## 2.1 $\hat{\beta}_{PPI,t}$ Updating Scheme

Here we describe the update scheme for $\hat{\beta}_{PPI}$ at time point $t$, which we denote $\hat{\beta}_{PPI,t}$. We shall assume here that $\Phi$ and $\Psi$ are known (with full generality in the complete paper). Based on our data generating equation, we have:

$$p(\beta_t | \boldsymbol{Y}^t) = p(\beta_t | \boldsymbol{Y}^t) \tag{4}$$

$$= p(\beta_t | y_{real,t}, y_{syn,t}, \boldsymbol{Y}^{t-1}) \tag{5}$$

$$\propto p(y_{real,t}, y_{syn,t} | \beta_t) p(\beta_t |, \boldsymbol{Y}^{t-1}) \tag{6}$$

$$= p(y_{real,t} | \beta_t) p(\beta_t |, \boldsymbol{Y}^{t-1}) * p(y_{syn,t} | \beta_t) p(\beta_t |, \boldsymbol{Y}^{t-1}). \tag{7}$$

$p(y_{real,t} | \beta_t)$ and $p(y_{syn,t} | \beta_t)$ are easy to sample from as we get them directly from the data generating equation. To estimate $p(\beta_t | \boldsymbol{Y}^{t-1})$ in an online fashion using a PPI estimator, we employ the dynamic structure used in McCormick et al. [2011] and assume that:

$$\beta_t | \hat{Y}_{t-1} \sim N(\hat{\beta}_{t-1}^{PPI}, \hat{\Sigma}_{t-1}^{PPI}) \tag{8}$$

$$\sim N(\hat{\beta}_{t-1}^{Naive} + \hat{\Delta}_{t-1}, \hat{\Sigma}_{t-1}^{Naive} + \hat{\Sigma}_{\Delta,t-1}). \tag{9}$$

This yields a Kalman filter based prediction equation:

$$\beta_t | \hat{Y}_{t-1} \sim N(\hat{\beta}_{t-1}^{PPI}, R_{t-1}^{PPI}) \tag{10}$$

where:

$$R_{t-1}^{PPI} = \hat{\Sigma}_{t-1}^{Naive} / \lambda^{Naive} + \hat{\Sigma}_{t-1}^{\Delta} / \lambda^{\Delta}$$

$\lambda^{Naive}$ and $\lambda^{\Delta}$ are fixed forgetting factors and are set to be less than 1. Estimation of the forgetting factors can be done via a model selection such as was done in McCormick et al. [2011].

To estimate $\hat{\beta}_{Naive,t}$ and $\hat{\Delta}_t$, we can use the fact that they are estimated on separate datasets which allows us to decompose the posterior into the product of two independent linear regression. Further improvements to estimating these terms can be done via the approach described in Hofer et al. [2024].

$$p(\hat{\beta}_{t-1}^{PPI}, \hat{\Delta}_{t-1} | \boldsymbol{Y}^{t-1}) = p(\hat{\beta}_{t-1}^{Naive} | Y_{syn}^{t-1}) p(\hat{\Delta}_{t-1} | Y_{real}^{t-1}). \tag{11}$$

### 2.1.1 Simulation

To demonstrate this updating scheme, we simulated a single iteration of the updating scheme described above. At time $t = 2$, we generated 5000 datapoints from the data generating process with $\beta = 1$, $x$ is a standard normal, and $B_\phi$ is a B-spline for x with 3 degrees of freedom using the **bs** function from the splines package in R R Core Team [2021] and $\Phi = [2, 4, -10]$. The upstream prediction model, $\hat{f}$ was set to be a simple linear regression with intercept. Note that this is purposefully a misspecified model and thus should not give unbiased estimates of $\beta$. At the next time step, $t = 3$, 20 real samples and 100 synthetic samples were generated from the data generating process and a forgetting factor of 0.0001 and 0.001 for $\lambda^{Naive}$ and $\lambda^\Delta$ respectively (the choice of very small values of forgetting factors is due to the large discrepancy between sample sizes and because in this simulation we are only doing one iteration. In a more realistic example with more steps, this discrepancy will be smaller and the forgetting factors will be much larger).

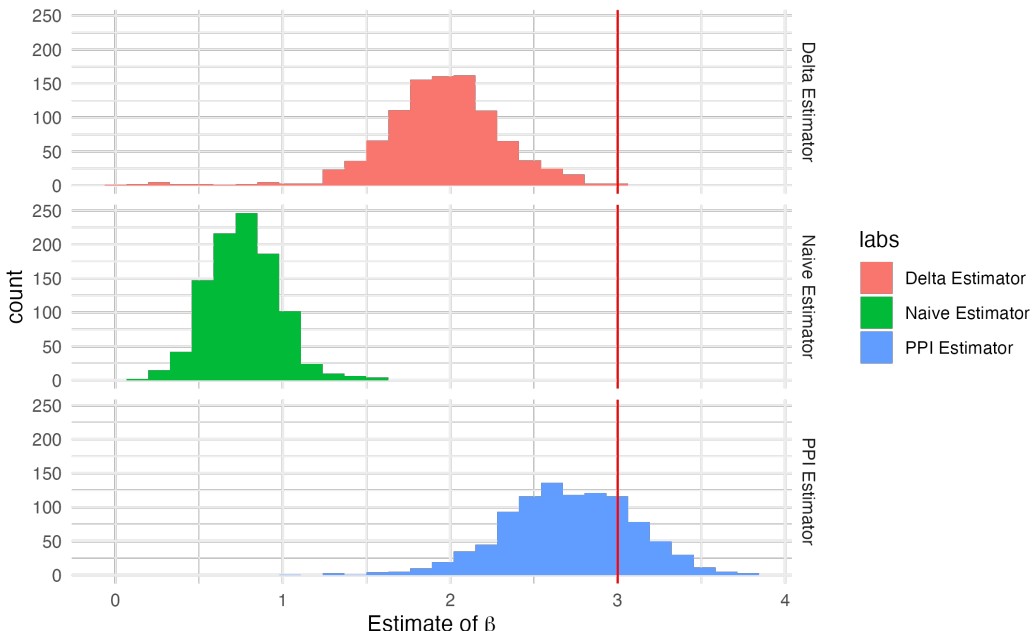

Figure 1: 1000 posterior draws from (top): $\hat{\Delta}$, (middle): $\hat{\beta}^{Naive}$, and (bottom): the PPI estimator. The vertical line at $\beta = 3$ represents the true value of $\beta$. Note that the PPI estimator is the only one that reasonably contains the true value of $\beta$. The slight negative bias in the PPI posterior distribution can be explained due to the choice of forgetting factor. More comprehensive choice of forgetting factors via model selection has the potential to ameliorate this issue.

Figure 1 illustrates 1000 posterior draws from $\hat{\Delta}$, $\hat{\beta}^{Naive}$ and $\hat{\beta}^{PPI}$. Because $\hat{f}$ was trained in the past and is misspecified, the posterior of $\hat{\beta}^{Naive}$ is very off, with a mean of 0.76 and standard error of 0.21. On the other hand, $\hat{\beta}^{PPI}$ is much closer to the true value of 3 with a mean of 2.7 and a standard error of 0.39, which leads to an overall lower MSE of 2.25 versus 0.491. This underestimation bias in our final estimator is because our proposed estimator is combining information from the parameter estimate in the past and the present. In the case of this simulation, at $t = 2$, $\beta$ was smaller. A comprehensive parameter tuning procedure to determine $\lambda^{naive}$ and $\lambda^\Delta$ has the potential to give even more accurate MSEs for $\beta^{PPI}$.

### 2.2 Sequential Experimental Design for $x_{real}$ and $x_{syn}$

A unique aspect of doing dynamic regression using IPD-based estimators is that one is performing inference using a combination of synthetic and real data. However, as noted in Hoffman et al. [2024b] as one diverges (either in time or in predictive accuracy) from the original upstream model, the optimal ratio of real and synthetic data to minimize the standard error of a PPI estimator changes.

To account for this, we propose at each time point to employ techniques from Bayesian optimal experimental design are used to determine the optimal ratio of real to synthetic data.

Specifically, we shall assume that at each time point, the researchers are able to spend their limited budget $C_t$ in two ways. They can spend it on collecting just $x$ values, each with a cost of $c_{syn}$ (which we shall refer to as "synthetic data") or spend $c_{real}$ and collect both $(x, y)$ (which we shall refer to as "real data").

To determine the best ratio of real to synthetic samples, we shall use techniques from Bayesian optimal experimental design Ryan et al. [2015], Lindley [1956]. A common setup in Bayesian optimal experimental design is to choose a design that maximizes the expected information gain at each time point. In our case, the space of our decisions are parameterized by $\zeta_t \in (0, 1)$ which is the fraction of budget, $C_t$ allocated to collecting real (each at cost $c_{real,t}$ with $C_t/c_{real,t}$ samples if the entire budget was spent on real data) and synthetic samples (each at cost $c_{syn,t}$ with $C_t/c_{syn,t}$ samples if the entire budget was spent on synthetic data). Following the setup, this yields a compact expression for estimating the Expected Information Gain (EIG) Rainforth et al. [2018] :

$$EIG(\zeta_t) = E_{\boldsymbol{Y}^t(\zeta_t),\beta}[H(p(\beta|\boldsymbol{Y}^{t-1}(\zeta_t))) - H(p(\beta|\boldsymbol{Y}^t(\zeta_t)))] \tag{12}$$

$$= E_{\boldsymbol{Y}^t(\zeta_t),\beta}[\log \frac{p(\boldsymbol{Y}^t|\beta)}{p(\boldsymbol{Y}^t)}]. \tag{13}$$

And Foster et al. [2019] illustrated that this double integral is well approximated via the nested-Monte Carlo approach of which each piece is estimable based on the derivations above:

$$EIG(\zeta_t) \approx \frac{1}{N} \sum_{i=1}^{N} \frac{p(\boldsymbol{Y}_n^t|\beta_{n,0})}{\frac{1}{M}\sum_{i=1}^{M} p(\boldsymbol{Y}_n^t|\beta_{n,m})}, \beta_{n,*} \sim p(\beta|\boldsymbol{Y}^{t-1}), \boldsymbol{Y}_n^t \sim p(\boldsymbol{Y}_n^t|\beta_{n,0})$$

Choosing, at each $t$, the sample size ratio $\zeta_t$ which maximizes our estimated EIG allows us to iteratively determine which type of sample–cheaper, but less reliable, synthetic data, or more expensive, but reliable, real data–would be most beneficial. An interesting point to note about this trade-off is that since, in the PPI estimator, the real samples help better estimate the bias of $\beta$ while the synthetic samples help estimate the variance of $\beta$, this procedure to balance between real and synthetic samples using $\zeta_t$ can also be thought of as an iterative bias-variance trade off.

## 3 Real Data Applications

Finally, to demonstrate the utility of this approach, we will apply this approach to two different application domains. In the first, we consider the problem of ocean going vessel fuel efficiency estimation. A major problem for shipping companies is that while many accurate physics-based models to estimate fuel use exist, due to issues such as wear and tear and barnacle buildup, the predictions from such models decay in accuracy the longer the ship is out of port Fan et al. [2022]. Thus, we should expect that such models are most useful close to port and progressively less useful the longer the ship has been out. By treating physics-model predicted efficiency as our synthetic data and ship-board sensor data as our real data, we can determine the optimal balance of the two to get the most accurate estimates of total fuel efficiency.

In the second context, we take a sociological problem of public opinion surveying. When designing such a survey, one must typically make a decision as to whether one would prefer the more expensive, but more accurate in-person surveys, or the cheaper but more prone to bias internet surveys Wu et al. [2022]. Inspired by Egami et al. [2023], we will demonstrate the optimal budget allocation between online and in-person polling to track public opinion on large studies with online and in-person components such as the American Community survey Bureau [2020].

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
