# OpenReview forum: "Bayesian Optimal Experimental Design of Streaming Data Incorporating Machine Learning Generated Synthetic Data"
_NeurIPS.cc/2024/Workshop/BDU — NeurIPS BDU Workshop 2024 Poster_

### Official Review · Reviewer_Vr51 · 2024-09-26
**Review of the online inference of model parameters and Bayesian optimal experiment design**

**Rating:** 4
**Confidence:** 5

**Review:**

This work addresses a very helpful direction of performing online statistical inference with streaming data and optimal ratio of real and synthetic data. It also discusses about several real-data applications. It adapts the methods so that it can perform Kalman filtering-based online inference.

However, I think there are a lack of enough data simulation and also missing real-data analysis results in the paper. For simulations, it only includes a set of simulation. For example, it didn't conduct enough simulation analysis including varying model parameters, sample sizes or forgetting factors.

Also for the part of optimal sample size ratio of real and synthetic data, it proposes the method but didn't have corresponding simulation analysis for the validation of the optimal size ratio decision.

It discusses about some real-data application examples, but the paper didn't conduct real-data analysis and demonstrates the performance of the proposed method.

Besides, I think a potential improvement may be to add priors for the forgetting rates and get the posterior inference of the rates.

---

### Official Review · Reviewer_Hw2B · 2024-10-03
**Review of Bayesian Optimal Experimental Design of Streaming Data Incorporating Machine Learning Generated Synthetic Data**

**Rating:** 7
**Confidence:** 3

**Review:**

The paper extends existing methods to perform inference incorporating synthetic data to the context of dynamic models. By leveraging prediction-powered inference estimators, the authors extend this approach to dynamic linear models and use optimal experimental design to iteratively determine the optimal amount of synthetic data to be incorporated into the model based on a budget constraint.

I found the paper interesting, and I believe that the topic is important.

---

### Decision · Program_Chairs · 2024-10-09

**Decision:**

Accept (Poster)

**Comment:**

Scores are mixed, one positive and one negative. The negative review is complaining about lack of comprehensiveness, but at the workshop stage I think this is a lesser criticism compared to the positive review, which comes from a seasoned expert. I therefore recommend acceptance.